# Torrefaction Upgrading of Heterogenous Wastes Containing Cork and Chlorinated Polymers

Andrei Longo [1], Catarina Nobre [2], Ali Sen [3], Roberta Panizio [2], Paulo Brito [2] and Margarida Gonçalves [1,2,*]

[1] Mechanical Engineering and Resource Sustainability Center, Department of Chemistry, Faculty of Sciences and Technology, NOVA University of Lisbon, Campus de Caparica, 2829-516 Caparica, Portugal

[2] VALORIZA—Research Centre for Endogenous Resource Valorization, Polytechnic Institute of Portalegre, Campus Politécnico 11, 7300-555 Portalegre, Portugal

[3] Forest Research Centre, School of Agriculture, University of Lisbon, Tapada da Ajuda, 1349-017 Lisboa, Portugal

* Correspondence: mmpg@fct.unl.pt

**Abstract:** Torrefaction of two mixed wastes composed of cork and chlorinated polymers was studied at temperatures from 200 to 350 °C, for residence times of 30 and 60 min. These wastes were recovered from sandwich panels with cork core, have different contents of cork biomass and chlorinated polymers and present poor fuel properties for energy recovery applications. The raw wastes and the produced biochars were characterized for proximate and ultimate analysis, chlorine content, mineral composition, calorific value, mass yield, energy density, particle size distribution, and adsorption capacity towards cationic and anionic dyes. Torrefaction enabled the production of biochars with mass yields from 97.2 to 54.5%, with an increase in 12.1 to 37.9% in apparent density relative to the raw wastes, and HHV from 18.2 to 20.7 MJ/kg. Nevertheless, the chlorine content of the biochars was increased to values higher than 5%, inadequate for solid fuels. Dechlorination of the biochars by washing with hot water enabled 84 to 91% removal of the chlorine species achieving final concentrations lower than 1%, without significant reduction in the biochars calorific values. For the waste with higher polymer and ash content, the torrefaction process reduced the heating value; therefore, energy valorization was not adequate. Both the raw wastes and the biochars were tested as adsorbents for cationic and anionic dyes. After activation with KOH, both the raw materials and the biochars had removal efficiencies higher than 90% for methylene blue, a cationic dye often found in industrial effluents. The results showed that torrefaction combined with hot water washing converted these wastes to biochars with the potential for energy or material valorization contributing to a circular economy in the cork industrial sector.

**Keywords:** torrefaction; energetic valorization; material valorization; industrial waste; polymeric waste; chlorine-containing waste

## 1. Introduction

Currently, the world faces serious problems of waste management related to the fast increase in the world population and associated waste production. These problems tend to worsen with the complexity of the wastes produced, which make it more difficult to find efficient management techniques [1]. The valorization of mixed wastes containing plastics is a specific challenge in industrial and municipal solid waste management due to their heterogeneity, unstable combustion, and harmful gas emissions when subjected to thermochemical valorization processes [2]. Furthermore, it is crucial to replace fossil resources with sustainable and renewable fuels and one possible pathway to achieve this goal is to encourage energy recovery from waste. Energy production from wastes can reduce negative environmental impacts caused by the excessive exploration and use of

fossil fuels whilst preventing the disposal of wastes in landfills, which is responsible for significant gaseous and liquid emissions that threaten public health [3]. This approach can reduce the demand for raw materials and, consequently, the environmental impacts of their extraction whilst decreasing dependence on fossil fuels by using wastes in energy recovery systems [4,5]. Portugal produced about 5.3 million tons of MSW and 11.4 million tons of IW in 2019. Approximately one-third of the produced MSW was landfilled and 80% of the IW was sent to recovery [6].

Thermochemical conversion processes, such as carbonization, pyrolysis, and gasification, are widely used to promote the energetic valorization of wastes. Besides reducing the amount of wastes disposed of in landfills, these processes can produce heat and waste-derived fuels that may be used in industrial processes, reducing energy and landfilling costs [7]. Nevertheless, there are some drawbacks regarding thermochemical valorization of mixed wastes such as municipal solid wastes or refuse-derived fuels, mainly due to their high heterogeneity and moisture content, low volumetric density, and poor grindability. Moreover, it is well-known that the polymeric fraction of wastes may create difficulties in the feeding systems of combustors or gasifiers, leading to equipment damage and increase in operational costs [8]. Furthermore, when the polymeric component contains chlorine, the formation of acid gas products during combustion or gasification can result in extensive corrosion of boilers and gasifiers [9].

Torrefaction is a mild thermochemical treatment carried out at temperatures between 200 and 300 °C in the absence of oxygen [10]. This process is used to improve the characteristics of biomass or wastes and convert them to high-quality solid fuels. After torrefaction, biomass or waste become more hydrophobic, reducing enzyme oxidation and minimizing decomposition by microorganisms [11]. Torrefaction significantly reduces the moisture and volatile content of the feedstock while increasing its heating value. The properties of the material produced after torrefaction depend on the nature of the biomass or waste used, as well as the torrefaction process conditions, i.e., temperature and residence time [12]. Due to this improvement of the physical and chemical characteristics of the torrefied material, this operation was used as pretreatment of biomass or waste before thermochemical conversion processes such as pyrolysis or gasification aiming to increase process efficiency [10]. Moreover, because of its increased brittleness, the torrefied material requires significantly lower energy expenses for grinding, thus reducing energy demand and costs of particle size reduction operations required before combustion or gasification [13].

Torrefaction has been studied to improve the physical and/or fuel properties of biomass and plastic-containing wastes. Barskov et al. [11] demonstrated the effectiveness of torrefaction in upgrading several lignocellulosic and non-lignocellulosic wastes such as agricultural wastes, food wastes, and sludge, converting them to value-added, energy products. Similar results were obtained by Niu et al. [10], who concluded that torrefaction at 250 °C can significantly improve the physical properties of biomass and make it more suitable for the gasification process. Dai et al. [14] showed that torrefaction of biomass before pyrolysis can improve fuel characteristics and promote economic benefits.

Even though torrefaction of several types of biomass have been studied recently, only a few works address the torrefaction of cork material [15,16]. Cork is a natural raw material obtained from the outer bark of cork oak (*Quercus suber*) and constitutes a renewable and biodegradable product with multiple industrial applications [17]. Cork is used in material applications ranging from cork stoppers, insulation or cladding panels, and adsorbent materials [18]. The production of cork-based materials generates a significant amount of cork waste which is usually recovered for cork powder applications or burned for electricity production since cork has a considerable high heating value and low ash content [19].

In the same context, torrefaction of MSW and refuse-derived fuel (RDF) has also been explored to increase homogeneity, reduce chlorine content, and improve fuel properties. Due to the high heterogeneity and presence of plastics, torrefaction of RDF/SRF requires more severe operating conditions [20]. The high chlorine content in polymeric wastes

containing PVC makes energetic valorization prohibitive due to harmful gas emissions and corrosion of the reactors and associated equipment [21].

Recari et al. [22] reported that the torrefaction of SRF as pretreatment produces biochars with better characteristics for gasification besides reducing the HCl content produced during the process. The reduction in chlorine content was also verified by Wang et al. [23] during the torrefaction of food waste. On the other hand, Nobre et al. [24] demonstrated the dichlorination of RDF samples through a torrefaction treatment followed by washing with hot water to remove ionic chlorine species and other water-soluble ash components.

The wastes used in this work were recovered from sandwich panels with a cork core and a plastic external layer (PVC) by mechanically cutting the panel some millimeters bellow the plastic layer to recover as much clean cork as possible. These wastes containing cork and plastic (PVC) are used as a fuel in ceramic kilns, but their heterogeneity leads to incomplete combustion causing the deposition of carbon impurities in the surface of the ceramic materials, thus lowering their quality. The high chlorine content of these wastes also limits their direct combustion or gasification due to corrosion effects related to acid gas products. This work aims to study torrefaction as a technology to reduce chlorine concentration and convert this waste into biochars with properties suitable for energy or material valorization. To the best of our knowledge this is the first study of thermochemical conversion of this type of waste and the results obtained may contribute to diversify the circular economy strategies within the cork industry.

## 2. Materials and Methods

### 2.1. Raw Materials

Two samples of industrial waste from the cork industry, CPW-1 and CPW-2, were provided by PRELIS—Smart Ceramics and are currently used as a fuel in ceramic kilns. These wastes are mainly composed of a mixture of lignocellulosic and polymer wastes from polymer-coated cork sheets. During their valorization, the cork is separated to be reused while the fraction adhered to the polymer (PVC) is ground into powder. This results in a residue composed of cork powder and PVC granules in which separation is not feasible.

The two samples used in this work differ in the proportion of polymers and ligno-cellulosic material, assessed through the selective dissolution method, as described by Ariyaratne et al. [25]. CPW-1 presented 17% of lignocellulosic material, 59% of polymers, and 24% of ash, while CPW-2 had approximately 49% of lignocellulosic material, 36% of plastics, and 15% of ash. The visual appearance of the two samples can be seen in Figure S1 (Supplementary Data).

### 2.2. Torrefaction Experiments

Torrefaction experiments were performed as described by Alves et al. [26]. Briefly, for each experiment, approximately 40 g of each sample were placed into a muffle furnace (Nabertherm® L3/1106, Lilienthal, Germany) using covered porcelain crucibles and heated at the desired torrefaction temperatures (200, 250, 300, and 350 °C) with residence times of 15, 30, and 60 min. After torrefaction, the chars were left to cool at room temperature in a desiccator and weighed using an analytical scale (Mettler Toledo AB204-S—0.1 mg, Columbus, OH, USA). Afterward, the chars were stored in plastic bags for further analysis.

### 2.3. Chemical Characterization and Fuel Properties

Before characterization analysis, CPW-1, CPW-2, and corresponding chars were milled (DeLongui mill, Treviso, Italy) and sieved (Retsch, Haan, Germany) to a particle size diameter <500 μm. All the determinations were conducted in triplicate, and the presented results correspond to average values.

Moisture, volatile matter (VM), and ash contents of the raw materials and corresponding chars were determined according to CEN/TS 15414-3:2010, EN 15402:2011, and EN 15403:2011, respectively. Fixed carbon (FC) content was obtained by difference on a dry basis (db). Ultimate analysis was carried out using an elemental analyzer (Thermo

Finnigan—CE Instruments Model Flash EA 112 CHNS series, Waltham, MA, USA) in order to assess the amount of nitrogen, carbon, hydrogen, and sulfur. Oxygen content was calculated by difference on a dry ash-free basis (daf). The high heating value (HHV) was measured using a calorimeter (IKA® C200, Staufen, Germany). Mineral composition (including chlorine content) of CPW-1, CPW-2, and corresponding chars were determined by X-Ray fluorescence (Niton XL3t XRF Analyzer, Waltham, MA, USA).

Particle size distribution is essential to deal with issues related to the transport and storage of wastes [20]. In this test, CPW-1 and CPW-2 chars produced with a residence time of 30 min were passed through three sieves (500, 250, and 125 µm) with agitation for 20 min to measure the mass of each fraction. The chars were not submitted to any milling process before the particle size distribution test.

Mass and energy yields of the produced chars (db) were calculated using Equations (1) and (2), respectively:

$$Mass\ yield\ (\%,\ db) = \frac{m_{char}}{m_{CPW}} \times 100 \qquad (1)$$

$$Energy\ yield\ (\%,\ db) = Mass\ yield \times \frac{HHV_{char}}{HHV_{CPW}} \times 100 \qquad (2)$$

where $m_{char}$ and $HHV_{char}$ are the mass and $HHV$ of the produced chars; $m_{CPW}$ and $HHV_{CPW}$ are the mass and $HHV$ of raw CPW samples. The energy yield is only focused on the amount of energy transferred from the raw CPW to each produced char.

The energy density of the produced chars (db) was calculated using Equation (3):

$$Energy\ density\ (db) = \frac{HHV_{char}}{HHV_{CPW}} \times 100 \qquad (3)$$

### 2.4. Thermal and Structural Analysis

The thermogravimetric analysis (TGA) of CPW-1, CPW-2, and corresponding chars was conducted using a Q50 TG (TA Instruments) thermogravimetric analyzer. The sample was heated from 30 to 950 °C at a constant rate of 10 °C min$^{-1}$ under a nitrogen atmosphere with a flow rate of 100 mL min$^{-1}$. The ignition and burnout temperatures of CPW-1, CPW-2, and corresponding chars were calculated using the intersection and conversion methods [27].

The Fourier Transform Infrared Spectroscopy (FT-IR) was carried out using a Nicolet iS10 FT-IR Spectrometer, Waltham, MA, USA, at wavenumbers ranging from 400 to 4000 cm$^{-1}$. Each sample was scanned 32 times, with a resolution of 4 cm$^{-1}$. Prior to analysis, samples were mixed with KBr powder at a sample: KBr ratio of 1 mg:100 mg. The sample-powder mixture was then reground with a mortar and pestle to ensure homogeneity. Pellets were created using 70–80 mg of powder in a pellet press at 10 MPa of pressure.

### 2.5. Char Washing for the Removal of Water-Soluble Chlorine

Upgrading of the chars by removal of water-soluble species was evaluated by washing the chars with hot distilled water. Briefly, the char samples of CPW-2 were mixed with distilled water in open glass beakers at an S/L ratio of 10 g/150 mL and heated to boiling point. After that, the samples were left to cool at room temperature and filtered. The washed chars were oven-dried (Memmert, Schwabach, Germany) at 105 ± 2 °C for 12 h. Chlorine content and HHV were measured in the decontaminated chars as described in Section 2.3.

### 2.6. Preliminary Adsorption Tests of the Produced Chars

CPW-1, CPW-2, and chars produced after 30 min of torrefaction at 250, 275, 300, and 350 °C were submitted to adsorption tests using two different dyes, methylene blue (MB) and methyl orange (MO) based on the work conducted by Correia et al. [28]. The choice of a cationic dye (MB) and an anionic dye (MO) aimed to evaluate the behavior of differently

charged species towards the biochars, namely, their interactions with the biochars surface charges and porous structures.

The samples were milled (Qilive mill—50 Hz) and sieved to particle size diameter of <500 μm (Retsch, Haan, Germany). After that, 50 mg of each sample (in triplicate) were placed in centrifuge tubes, and 5 mL of MB or MO aqueous solution (100 mg L$^{-1}$) was added. The tubes were shaken for 10 s (Heidolph Reax top, Darmstadt, Germany), and after the contact time, the mixture was centrifuged at 3500 rpm for 10 min (Hettich EBA 20, Sigma-Aldrich, Burlington, MA, USA). The absorbance of the supernatant was measured through UV/Vis spectroscopy (Biochrom Libra S4, Tuttlingen, Germany) using as maximum wavelengths 668 nm for MB and 464 nm for MO. The same procedure was used to assess the effect of contact time after 1 week for both dyes. Moreover, to assess the potential of enhancing adsorption capacity, the produced chars were chemically activated with KOH, as described by Regmi et al. [29], and submitted to the adsorption test conditions described above. Removal efficiency (*R*) was calculated using Equation (4):

$$R(\%) = \frac{C_0 - C_f}{C_0} \times 100 \tag{4}$$

where $C_0$ and $C_f$ are the initial and final dye concentrations, respectively.

## 3. Results and Discussion

### 3.1. Raw Material Characterization

The results of the selective dissolution test showed that CPW-1 is mainly composed of polymers while CPW-2 presented a higher proportion of cork (Figure 1). These differences are related to the efficiency of the recycling process that may recover a thinner or thicker layer of cork close to the external polymer layer thus producing CPW with different proportions of cork biomass and polymer.

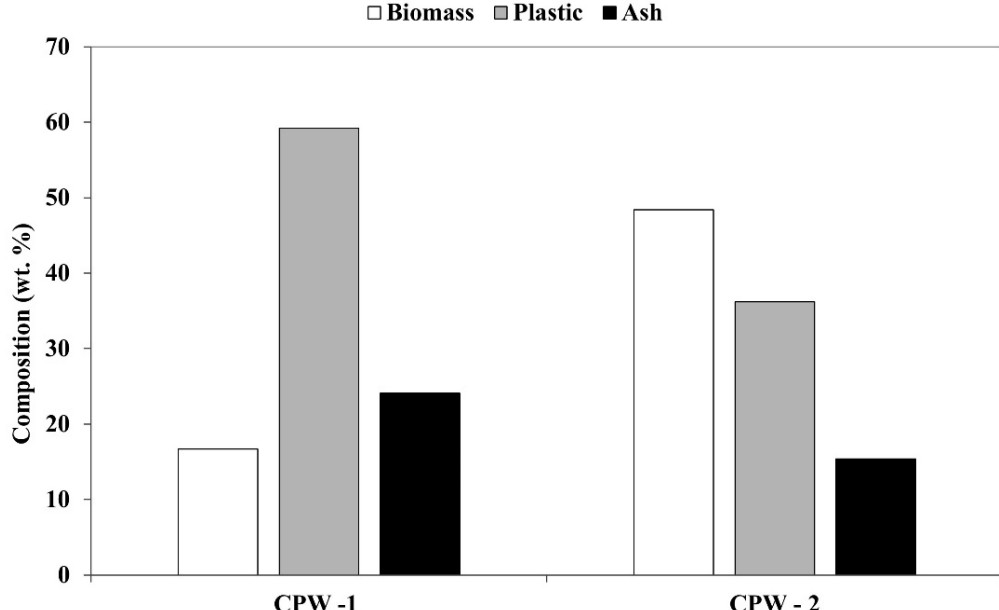

**Figure 1.** Relative composition of CPW-1 and CPW-2 as determined by the selective dissolution test.

The higher polymer content of CPW-1 is expected to positively affect its carbon content but can also cause aggregation problems and release acid compounds during thermal treatments [21].

Both wastes had a low initial moisture content representing 4.8 wt.% in mass for CPW-1 and 6.1 wt.% for CPW-2 (Table 1). These low values are similar to values presented in other works concerning cork powder wastes, mainly related to the low permeability

of cork due to its high suberin content [30]. The volatile matter was higher for CPW-2 because of its higher proportion of biomass compared with CPW-1. Regarding the amount of ash present in both wastes, CPW-1 (24 wt.%) showed higher ash content compared with CPW-2 (15 wt.%). These values reflect the ash content of cork biomass [31] and the presence of mineral additives (fillers) incorporated in the PVC structure [32]. Fixed carbon was relatively low for both wastes (7.6 for CPW-1 and 6.3% for CPW-2) as a consequence of their high ash and volatile matter contents.

**Table 1.** Raw material characterization (proximate and ultimate analysis, mineral composition, and HHV).

| | CPW-1 | CPW-2 |
|---|---|---|
| Proximate analysis (wt.%) | | |
| Moisture | 4.8 ± 0.04 | 6.1 ± 0.07 |
| Volatile matter [a] | 68.3 ± 1.2 | 78.3 ± 1.2 |
| Ash [a] | 24.1 ± 0.6 | 15.4 ± 1.7 |
| Fixed carbon [a] | 7.6 | 6.3 |
| Ultimate analysis (wt.%) | | |
| C | 42.6 | 46.2 |
| H | 5.6 | 5.8 |
| O [b] | 27.1 | 29.3 |
| N | 0.6 | 3.3 |
| S | 0.0 | 0.0 |
| Mineral composition (mg/g) | | |
| Ca | 102.3 ± 0.3 | 85.4 ± 0.2 |
| K | 2.4 ± 0.09 | 0.5 ± 0.07 |
| Fe | 0.7 ± 0.03 | 0.2 ± 0.02 |
| Ti | 0.5 ± 0.01 | 0.4 ± 0.01 |
| Si | 2.0 ± 0.7 | 0 ± 0.9 |
| Zn | 0.3 ± 0.01 | 0.1 ± 0.01 |
| Sc | 0.2 ± 0.03 | 0.3 ± 0.03 |
| Cl | 71.8 ± 0.4 | 32.5 ± 0.3 |
| Others | 0.2 | 0.3 |
| Fuel properties | | |
| O/C | 0.64 | 0.63 |
| H/C | 0.13 | 0.12 |
| HHV (MJ kg$^{-1}$) | 17.2 | 17.6 |

[a] Dry basis; [b] by difference, O (%) = 100-C-H-N-S.

The results from the elemental analysis indicate that both wastes had substantial amounts of carbon, specifically 42.6% for CPW-1 and 46.2% for CPW-2, and hydrogen contents of 5.6% for CPW-1 and 5.8% for CPW-2. Thus, the H/C and O/C ratios showed approximate values for both wastes (0.13 and 0.12 for H/C and 0.64 and 0.63 for O/C to CPW-1 and CPW-2, respectively). The presence of chlorine in the PVC explains why the carbon content of CPW-1 is comparable to CPW-2 regardless of its higher polymer content.

These composition characteristics are coherent with the similar HHV values obtained for both wastes (17.2 and 17.6 MJ/kg). Nitrogen content was slightly higher for CPW-2 (3.3%), mainly due to the higher amount of lignocellulosic materials, and sulfur was absent in both samples. Low values for nitrogen and sulfur in the raw material are important parameters regarding $NO_x$ and $SO_x$ emissions and, consequently, contribute to reducing the environmental impacts from the waste valorization chain.

Results from ash mineral composition showed that Ca and Cl are the major elements found in the ash from both wastes. Together, they represent about 96.5% for CPW-1 and approximately 98.4% for CPW-2. High Ca concentration is expected due to the mineral composition of cork and the fillers used during PVC production, while Cl may come from

the composition of the polymeric fraction [32]. The high Cl content in both wastes suggests that a pretreatment for dechlorination might be required before energy valorization of such wastes. The release of Cl during the thermochemical processes can corrode equipment, increase costs with maintenance, and consequently reduce process efficiency [9,21]. The sum of other elements in small concentrations, including K, Fe, Ti, Si, Zn, and Sc, represents 3.5% and 1.6% for CPW-1 and CPW-2, respectively. Other elements such as Sr, Cu, Sn, and Cd were found in trace concentrations, and their sum was less than 0.5%.

The amount of ash and its composition are two important parameters to assess the feasibility of the energetic or material valorization of wastes. High ash content may be non-profitable for energetic purposes, while the presence of alkaline and alkaline earth metals on ashes may bring issues related to slagging and fouling in gasifiers during energetic valorization processes [10,33].

### 3.2. Torrefaction Experiments

Increasing torrefaction conditions leads to the production of darker and denser chars (Figure S2, Supplementary Data) as well as a significant increase in grindability, in line with the observations of other works regarding torrefaction of biomass and wastes [20,34]. In some cases, even if the produced chars are not profitable for energy or material recovery, the sharp volume reduction can ease logistics and reduce costs related to landfill disposal (Figure 2).

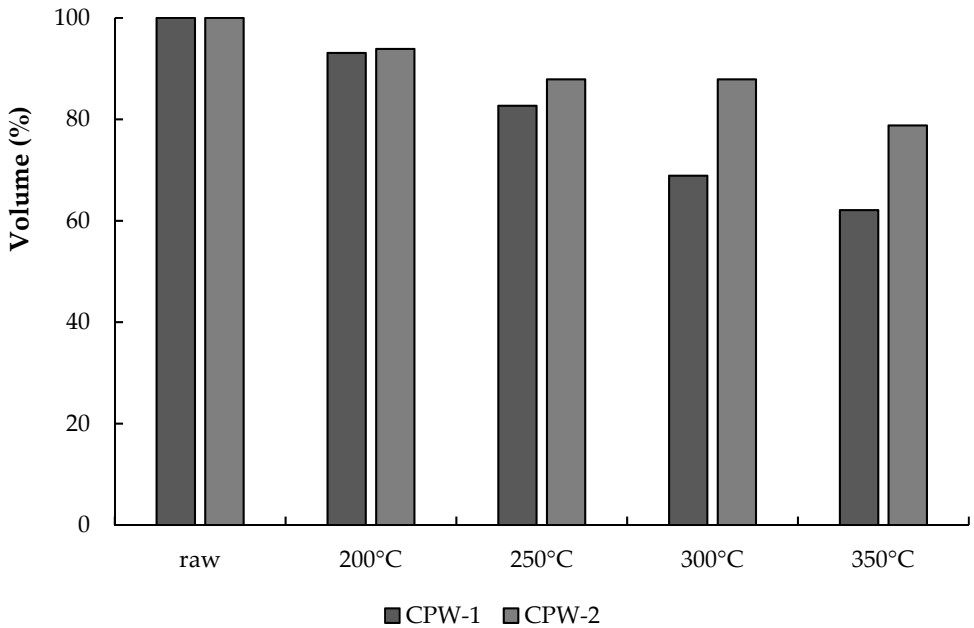

**Figure 2.** Volume reduction for CPW-1 and CPW-2 after torrefaction for 60 min.

One of the benefits of waste torrefaction before its valorization is the observed volume reduction, which leads to an increase in density. This physical property has implications on the decision-making process for waste valorization because wastes with higher bulk density are easier to handle, transport, and store, increasing the efficiency of the process and reducing costs [35]. Biochars produced in this study had an increase in apparent density from 12.1 to 37.9% and these results are in accordance with values for biochars obtained from different raw materials that show an increase in apparent density in the range of 13 to 55.6% for lignocellulosic and refuse-derived fuel wastes, respectively [20,36]. According to Figure 2, the volume reduction was positively correlated to the torrefaction temperature reflecting the transition from the lignocellulosic and polymeric structures to the more aromatic carbonaceous structures of the biochars [37]. This effect was more evident for CPW-1, composed mainly of polymers that have typically low apparent densities [38]. Transportation and landfilling represent large environmental impacts due to the

high consumption of fossil fuels and release of $NO_x$ and methane–air emissions coupled with heavy metal water emissions, respectively [39]. As such, this volume reduction via torrefaction seems to be an attractive way to significantly reduce the costs and increase the sustainability of waste valorization. Moreover, when waste valorization is not possible, landfilled wastes would require a smaller area.

Mass Yield, HHV, Energy Yield, and Energy Density

As expected, char yield was negatively correlated with torrefaction temperature and residence time. Increasing torrefaction severity leads to a decrease in the dry solid mass yield ranging from 97.2 to 94.9% in the mildest torrefaction condition (200° C/15 min) to 62.7 and 54.5% for the more severe condition (350 °C/60 min) for CPW-1 and CPW-2, respectively (Table 2). A similar trend for mass reduction was found after torrefaction experiments with RDF and lignocellulosic wastes [20]. Because decomposition of the polymeric fraction requires higher temperatures and longer residence time, the mass reduction was higher for CPW-2, composed mainly of cork wastes.

**Table 2.** Mass yield, HHV, and energy yield for CPW-1- and CPW-2-produced chars.

| Process Conditions | | CPW-1 | | | | CPW-2 | | | |
| --- | --- | --- | --- | --- | --- | --- | --- | --- | --- |
| T (°C) | t (min) | Mass Yield (%) | HHV (MJ kg$^{-1}$) | Energy Yield (%) | Energy Density | Mass Yield (%) | HHV (MJ kg$^{-1}$) | Energy Yield (%) | Energy Density |
| Raw | — | 100 | 17.2 | 100 | 1.00 | 100 | 17.6 | 100 | 1.00 |
| 200 | 15 | 97.2 | 15.3 | 86.6 | 0.89 | 94.9 | 18.0 | 97.5 | 1.03 |
| | 30 | 96.6 | 15.7 | 88.2 | 0.91 | 93.7 | 18.2 | 96.9 | 1.03 |
| | 60 | 96.6 | 14.9 | 83.6 | 0.87 | 92.9 | 18.0 | 95.3 | 1.03 |
| 250 | 15 | 96.1 | 16.0 | 89.3 | 0.93 | 92.9 | 18.3 | 96.9 | 1.04 |
| | 30 | 94.6 | 15.6 | 86.1 | 0.91 | 85.6 | 19.1 | 93.0 | 1.09 |
| | 60 | 84.7 | 16.5 | 81.6 | 0.96 | 79.9 | 19.8 | 89.9 | 1.13 |
| 300 | 15 | 89.5 | 15.1 | 79.1 | 0.88 | 85.0 | 19.0 | 91.9 | 1.08 |
| | 30 | 77.8 | 16.0 | 72.5 | 0.93 | 66.3 | 19.1 | 72.2 | 1.09 |
| | 60 | 69.6 | 14.1 | 57.2 | 0.82 | 60.2 | 20.5 | 70.3 | 1.17 |
| 350 | 15 | 71.3 | 14.9 | 62.1 | 0.87 | 60.1 | 20.5 | 70.1 | 1.17 |
| | 30 | 63.9 | 14.4 | 53.7 | 0.84 | 55.3 | 20.7 | 65.3 | 1.18 |
| | 60 | 62.7 | 14.3 | 52.4 | 0.83 | 54.5 | 18.6 | 57.8 | 1.06 |

At 200 °C, the decrease in mass yield possibly occurs mainly by the degradation of hemicellulose but also by the decomposition of lignin [37]. Increasing the temperature to 250 °C initiates the degradation of suberin [40]. As suberin is the most abundant component in cork, the reduction in mass yield can be clearly observed when the torrefaction temperature is 300 °C or higher. Although residence time also influences the decrease in mass yield, especially at higher temperatures, the temperature seemed to have a greater effect on mass reduction [10,13].

Low torrefaction temperature (200 °C) did not have a substantial effect on improving the fuel properties of the chars compared with the raw materials because PVC, cellulose, and lignin are relatively stable at that temperature [41]. Visual appearance, ash content, mass yield, and HHV had small changes at this temperature for all residence times. On the other hand, high torrefaction temperatures (300–350 °C) can substantially enhance the fuel characteristics, such as HHV and energy density. Nevertheless, the most severe torrefaction conditions (350 °C/60 min) led to a sharp increase in ash content and a consequent decrease in HHV for CPW-1 and CPW-2 chars.

HHV showed the opposite behavior between the two samples as the torrefaction severity increased. For CPW-1, increasing torrefaction conditions led to an HHV decrease, with the highest value obtained for the raw material sample (17.2 MJ kg$^{-1}$) and decreasing

progressively as torrefaction severity conditions increased. The lowest values for this parameter were observed at 300 °C/60 min (14.1 MJ kg$^{-1}$). Harsh torrefaction conditions lead to an increase in ash percentage and, consequently, reduced HHV [42].

On the other hand, HHV increased on CPW-2 chars for all tested conditions compared with the raw material. HHV increased approximately 8.5% at 250 °C/30 min, and 17.6% at 350 °C/30 min while mass yields were of 85.6% and 55.3%, respectively, at those temperatures. Considering the physical improvements, this trend can indicate that torrefaction has good potential to be used as a pretreatment before the energy valorization of this waste. For the most severe torrefaction conditions (350 °C/60 min), HHV increase was less significant due to the more extensive release of carbon components and volatile matter, leading to a high ash concentration on this char sample.

Energy yield represents the amount of energy of the raw material retained in the solid product. As seen in Table 2, CPW-1 chars showed overall lower energy yields than CPW-2 chars. Values above 80% were observed for chars produced at 200–250 °C for all residence times and for both wastes. When the temperature increased to 300 °C, the energy yield was 79.1 and 91.9% for CPW-1 and CPW-2, respectively, for a residence time of 15 min. Beyond this condition, increasing temperature or residence time leads to a rapid decrease in energy yield, reaching minimum values at the most severe torrefaction condition (350 °C/60 min.).

Another advantage of the torrefaction process is the increase in the energy density of the biochars compared with the raw material. This is generally a consequence of the loss of components that do not contribute to the heating value (such as oxygen or nitrogen) and accumulation of carbon and hydrogen in the biochars [41]. This increase in the energy density of the biochars depends on the waste composition, torrefaction conditions, and reactor type [10]. These factors also determine the reduction in the energy yield that tends to decrease with the increased severity of torrefaction conditions. CPW-2 showed a gradual increase in energy density, which ranged from 1.02 to 1.18, justifying the torrefaction process as pretreatment to improve the combustible characteristics of the raw material. On the other hand, CPW-1 showed a reduction in energy density with the increase in torrefaction conditions showing values between 0.82 and 0.96 (Figure S3—Supplementary Data). The lower energy yield and energy density of this sample is justified by the higher concentration of polymers, and in particular PVC which decomposes at that temperature yielding predominantly gas products, and therefore having poor carbon retention in the biochars [43].

According to these results, mild and severe torrefaction (250–350 °C) seem to have a great potential to be applied in CPW-2 as a pretreatment in energy valorization processes since its combustible characteristics such as energy density and HHV are enhanced and can compensate the mass loss.

*3.3. Char Characterization*

3.3.1. Particle Size Distribution

Particle size distribution showed a different pattern between both samples (Figure 3). As CPW-1 has higher amounts of polymers, increasing torrefaction conditions lead to a particle size increase, mainly due to agglomeration caused by the melting of the polymeric fraction of the samples. As polymers melt during thermal treatment, the fine particles become stuck in this material and produce larger particles. This behavior is more evident in more severe torrefaction conditions, and larger particles were observed for the char produced at 350 °C for 60 min.

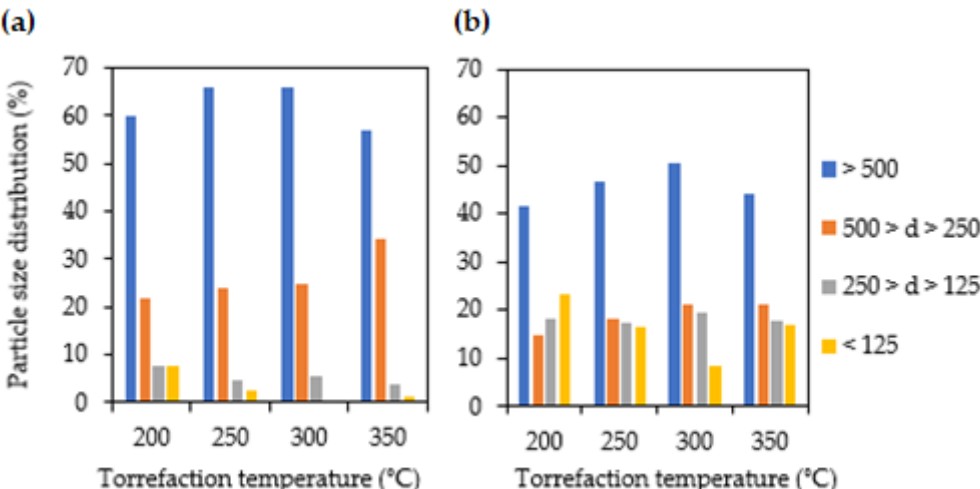

**Figure 3.** Particle size distribution for chars produced between 200 and 350 °C with a residence time of 30 min. (**a**) CPW-1 and (**b**) CPW-2.

The results for CPW-2 showed the opposite trend. At more severe torrefaction conditions, chars presented a higher proportion of fine particles compared with CPW-1, enforcing the influence of the polymeric fraction during torrefaction. Even though the particles become larger after torrefaction in CPW-1, both wastes became more brittle after torrefaction, which has the potential to reduce energy costs during grinding processes [10]. This reduction in particle size is also a relevant parameter for combustion or gasification since it influences heat transfer and contact with the oxidizing agent [44].

### 3.3.2. Proximate and Ultimate Analysis

At milder temperatures, the moisture content constantly decreased, reaching lower values at 200 °C for 30 min for CPW-1 (1.4%) and 250 °C for 15 min for CPW-2 (1.7%). The raising of torrefaction severity leads to an unexpected increase in the moisture content of the chars (Table 3). This may occur when the raw materials generate a large proportion of volatile products that are adsorbed in the biochar structure and are eliminated during the moisture determination by heating overnight at 105 °C. This behavior, which is not usual for lignocellulosic materials, is more frequent during carbonization of polymeric wastes, due to the high volatility of the oligomeric decomposition products [45].

Compared with the raw materials, the biochars presented lower moisture and volatile matter contents and higher fixed carbon and ash contents, as expected, since it is a process that tends to eliminate, as gases or tars, the decomposition products that are volatile at the carbonization temperature. Volatile matter decreased with increasing temperature and residence time reaching the lowest values for the biochars produced at 350 °C from CPW-1 (51.1%) and from CPW-2 (47.7%).

As expected, ash content increased gradually as torrefaction conditions became more severe. Ash content of the produced chars ranged from 23.1 to 41.3% for CPW-1 and 15.2 to 30.8% for CPW-2. These values are very high compared with raw cork powder wastes, which have an ash content of approximately 3% [31]. For the wastes analyzed in this study, ash content was already high (24.1% and 15.4% for CPW-1 and CPW-2, respectively) which may be the result of some contamination during the waste processing process and the contribution of mineral fillers that are used during the PVC production process [46].

**Table 3.** Proximate analysis for CPW-1 and CPW-2 chars.

| Process Conditions | | CPW-1 | | | | CPW-2 | | | |
|---|---|---|---|---|---|---|---|---|---|
| | | Moisture (%) | VM (%) | Ash (%) | FC (%) | Moisture (%) | VM (%) | Ash (%) | FC (%) |
| Raw material | | 4.8 | 68.3 | 24.1 | 7.6 | 6.1 | 78.3 | 15.4 | 6.3 |
| T (°C) | t (min) | | | | | | | | |
| 200 | 15 | 2.2 | 67.5 | 23.1 | 9.4 | 2.6 | 75.1 | 15.2 | 9.7 |
| | 30 | 1.6 | 69.4 | 23.3 | 7.3 | 2.1 | 72.3 | 17.6 | 10.1 |
| | 60 | 1.4 | 68.6 | 27.9 | 3.5 | 2.0 | 72.3 | 15.9 | 11.8 |
| 250 | 15 | 1.6 | 69.3 | 24.1 | 6.6 | 1.7 | 74.6 | 15.8 | 9.6 |
| | 30 | 1.7 | 68.4 | 25.0 | 6.6 | 2.7 | 70.4 | 17.2 | 12.4 |
| | 60 | 4.1 | 63.3 | 27.3 | 9.4 | 2.9 | 64.8 | 17.2 | 18.0 |
| 300 | 15 | 2.9 | 65.3 | 26.5 | 8.2 | 2.4 | 68.3 | 17.8 | 13.9 |
| | 30 | 3.2 | 58.3 | 28.7 | 13.0 | 3.7 | 55.2 | 23.2 | 21.6 |
| | 60 | 4.7 | 55.3 | 34.8 | 9.9 | 4.9 | 48.0 | 23.7 | 28.3 |
| 350 | 15 | 3.0 | 57.5 | 31.6 | 10.9 | 4.1 | 51.6 | 24.5 | 23.9 |
| | 30 | 3.2 | 52.6 | 34.6 | 12.8 | 4.5 | 47.7 | 25.0 | 27.3 |
| | 60 | 3.2 | 51.1 | 41.3 | 9.6 | 5.2 | 49.4 | 30.8 | 19.8 |

On the other hand, even though fixed carbon concentration is an important parameter to assess the HHV of fuels, high ash concentration can be a challenge when the goal is energy recovery from wastes. It may be concluded due to the high proportion of Ca present in ashes that these fillers are composed mainly of $CaCO_3$ [47]. Thus, assessing initial waste composition through proximate analysis is a crucial step to decide the most suitable way to promote waste recovery.

The torrefaction process significantly increased the carbon content of the biochars relatively to the raw materials and that effect was positively correlated with the torrefaction temperature (Table 4). Moreover, the O/C and H/C ratios decreased with the increasing process temperature for both wastes due to the elimination of oxygen through the decarboxylation and decarbonylation reactions. The chars produced at 350 °C for 30 min of residence time showed a considerable reduction in O/C ratio (67% for CPW-1 and 32% for CPW-2) and H/C (38% for CPW-1 and 42% for CPW-2), achieving values comparable to solid fuels such as bituminous char or lignite [24]

One of the greatest obstacles to the thermochemical valorization of polymeric material wastes containing PVC is the presence of chlorine. The chlorine present in this sort of polymer is released in the form of gases (i.e., HCl, dibenzodioxins, and dibenzofurans) when submitted to higher temperatures that can cause corrosion of the reactors or gasifiers, turning the recovery process unfeasible for economic reasons [10]. Moreover, dibenzodioxins and dibenzofurans are extremely harmful to the environment and, depending on their concentration in the raw material, it can become impeditive to their valorization by thermochemical methods [48].

Due to the polymeric fraction present in these wastes, CPW-1 and CPW-2, chlorine showed high concentrations in the original feedstock, 7.3 and 3.4%, respectively. It represents approximately 71.8 and 32.5 mg/g for CPW-1 and CPW-2, respectively. The torrefaction process at temperatures higher than 300 °C can break the bonds between carbon and chlorine, forming inorganic $Cl^-$ [49] that tends to be retained in the biochar as it happens to other mineral components [20]. Therefore, the chlorine content of the biochars increased with the carbonization temperature (250 °C for CPW-1 and 300 °C for CPW-1) and then decreased for higher temperatures probably because of the release of organochlorine compounds during the pyrolytic decomposition of the biochars [50]. These results suggest that a pretreatment of dechlorination of the chars produced must be carried out in order to reach chlorine concentrations compatible with their thermochemical valorization.

**Table 4.** Ultimate analysis and ash mineral composition for chars produced after torrefaction (30 min).

| | CPW-1 | | | | CPW-2 | | | |
|---|---|---|---|---|---|---|---|---|
| | **200 °C** | **250 °C** | **300 °C** | **350 °C** | **200 °C** | **250 °C** | **300 °C** | **350 °C** |
| | Ultimate analysis (wt.%, daf) | | | | | | | |
| C | 58.8 | 54.9 | 61.3 | 76.7 | 52.7 | 51.8 | 58.3 | 64.4 |
| H | 6.5 | 5.0 | 5.8 | 6.2 | 6.4 | 4.6 | 4.5 | 4.6 |
| N | 1.2 | 0.5 | 0.5 | 0.7 | 3.1 | 2.8 | 3.0 | 3.6 |
| S | 0 | 0.2 | 0 | 0 | 0 | 0 | 0 | 0 |
| O | 33.5 | 39.4 | 32.4 | 16.4 | 37.7 | 40.8 | 34.2 | 27.4 |
| H/C | 0.11 | 0.09 | 0.09 | 0.08 | 0.12 | 0.08 | 0.08 | 0.07 |
| O/C | 0.57 | 0.72 | 0.53 | 0.21 | 0.72 | 0.79 | 0.59 | 0.43 |
| | Ash mineral composition (mg/g) | | | | | | | |
| Ca | 117.1 ± 0.3 | 96.5 ± 0.3 | 132.2 ± 0.3 | 181.7 ± 0.4 | 53.7 ± 0.2 | 73.6 ± 0.2 | 119.4 ± 0.3 | 119.6 ± 0.3 |
| K | 1.9 ± 0.1 | 1.5 ± 0.07 | 0.2 ± 0.08 | 1.4 ± 0.09 | 0.8 ± 0.06 | 0.8 ± 0.07 | 0.6 ± 0.09 | 1.0 ± 0.09 |
| Fe | 1.2 ± 0.04 | 0.4 ± 0.02 | 0.9 ± 0.05 | 0.6 ± 0.03 | 0.2 ± 0.02 | 0.2 ± 0.02 | 0.3 ± 0.02 | 0.6 ± 0.05 |
| Ti | 0.4 ± 0.02 | 0.5 ± 0.01 | 0.6 ± 0.03 | 1.9 ± 0.03 | 0.3 ± 0.01 | 1.2 ± 0.02 | 2.0 ± 0.04 | 1.6 ± 0.04 |
| Si | 3.3 ± 0.7 | 0 ± 1.1 | 2.4 ± 0.8 | 2.5 ± 0.7 | 0 ± 0.7 | 1.3 ± 0.6 | 0 ± 1.2 | 2.3 ± 0.9 |
| Zn | 0.3 ± 0.01 | 0.3 ± 0.01 | 0.5 ± 0.01 | 0.6 ± 0.01 | 0.1 ± 0.01 | 0.2 ± 0.01 | 0.3 ± 0.01 | 0.3 ± 0.01 |
| Sc | 0.2 ± 0.03 | 0.2 ± 0.03 | 0.4 ± 0.03 | 0.5 ± 0.04 | 0.2 ± 0.02 | 0.4 ± 0.02 | 0.4 ± 0.03 | 0.5 ± 0.03 |
| Cl | 56.5 ± 0.4 | 100.3 ± 0.6 | 85.2 ± 0.6 | 50.2 ± 0.4 | 15.8 ± 0.2 | 53.8 ± 0.4 | 71.7 ± 0.5 | 60.2 ± 0.5 |
| Others | 0.3 | 0.1 | 2.8 | 0.2 | 0.8 | 3.2 | 2.3 | 4.1 |

### 3.3.3. Chlorine Content and HHV after Washing Process

The removal of chlorine from the CPW-2 biochars was evaluated by washing with hot water, a process that promotes solubilization of the inorganic chlorine species.

The washing process reduced the chlorine content of the raw material and the biochar obtained at 200 °C, but chlorine removal efficiency was below 25% and corresponded to the solubilization of inorganic chlorine species present in the biomass fraction. As for the biochars, initial chlorine concentrations were higher than 5% and chlorine removal efficiencies between 84 and 91% were obtained after washing with hot water, reaching final chlorine concentrations lower than 1%, a value adequate for solid fuels for use in boilers or gasifiers [21]. These results indicate that torrefaction at temperatures above 250 °C concentrate the chlorine present in the raw material in the biochar product, but in the form of inorganic chlorine that may be removed by washing with hot water [24]. The reduction was more evident in chars produced at higher temperatures, reaching 91% chlorine removal for the chars produced at 350 °C. Nonetheless, maximum chlorine concentration in the char was observed at 300 °C, indicating that some chlorine was eliminated with the gas products and tars during torrefaction at 350 °C.

The HHV of the raw material (CPW-2) and of the biochars obtained at the different temperatures were determined before and after the washing process to evaluate the effect of this process in the energy content of the produced biochars. The results demonstrated little decrease in this parameter after the washing process, indicating that most carbon species remain in the carbonaceous structure. The calorific value of CPW-2 was reduced by 17.5% after the washing process, reflecting the dissolution of some organic compounds in the aqueous phase. For the biochars produced at temperatures from 200 °C to 300 °C the HHV reduction after washing varied from 12.5 to 1.3%, reflecting the growing hydrophobicity of the biochars. For the chars produced at 350 °C, HHV reduction after washing was of 14%, probably because at this temperature, some pyrolytic degradation of the biochar occurs, producing low molecular weight species that remain adsorbed in the char but may be leached by the hot water washing.

For the chars produced at 250 and 300 °C, the HHV was 19 MJ/kg before the washing and decreased to 18.2 and 18.9 MJ/kg, respectively, values that are compatible with the energy valorization of those chars [11].

### 3.3.4. Thermal Degradation Behavior

The thermal degradation behaviors of wastes are of interest in thermal processes because process conditions such as temperature and solid residence times can be optimized using this information. The thermal degradation of cork is well-known through scanning electron microscopy and thermogravimetric analyses [51]. Cork undergoes thermal degradation at temperatures above 200 °C, and the magnitude of the thermal degradation depends on the final temperature, heating rate, chemical, and anatomical compositions of cork [52]. However, when biomass is blended with polymers, its thermal degradation behavior is altered depending on the used polymer types [38]. The thermal degradation patterns of CPW-1 and CPW-2 are shown in Figure 4.

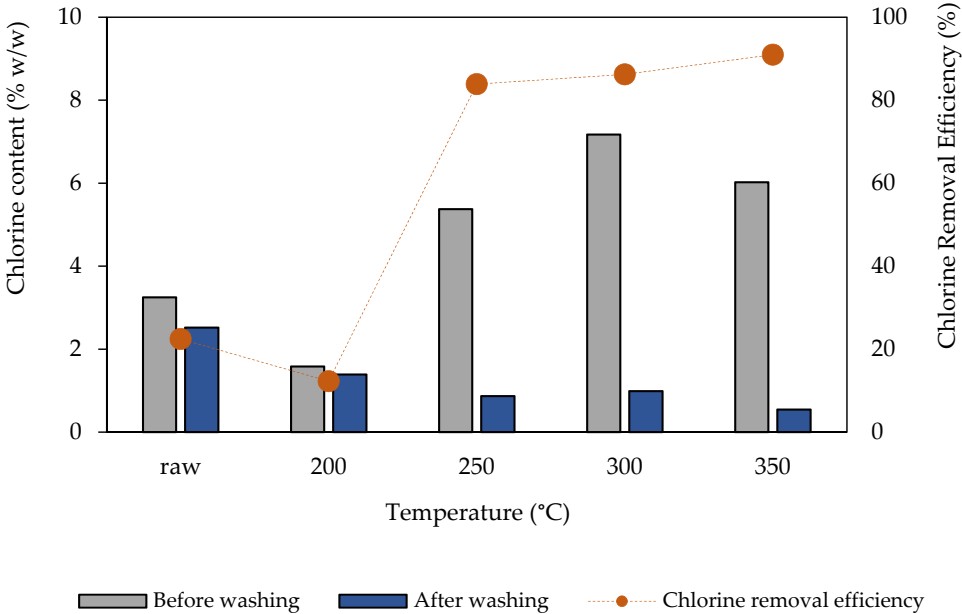

**Figure 4.** Chlorine content of the raw material and the biochars before and after the washing process.

As seen in Figure 5a, torrefaction treatment increased the thermal stabilities of the cork wastes. For instance, at 750 °C, the raw waste lost 80% of its mass while the chars obtained at 350 °C suffered a 50% mass loss, while raw cork waste showed about 80% mass loss. The thermal degradation of this type of mixed waste involves devolatilization and char decomposition reactions [40]. As can be seen from Figure 5b, cork-polymer wastes undergo a devolatilization reaction at temperatures of approximately 310 °C and two char decomposition reactions at temperatures between 440 and 800 °C. Dehydrochlorination reactions take place along with biomass devolatilization reactions [53]. The devolatilization reactions occur at a higher rate in raw cork-polymer wastes or lower temperature chars due to their higher volatile content. Interestingly, the two cork-polymer wastes showed similar thermal decomposition patterns indicating that after carbonization both exhibit a homogeneous carbonaceous structure (Figure 5c). The waste chars also showed similar degradation rates but their devolatilization and char decomposition patterns varied. The peak temperatures of the first and second char decomposition reactions occurred at slightly higher temperatures for CPW-2 and derived chars. The overall results indicate that maximum thermal degradation temperatures range between 750 and 800 °C for cork-polymer wastes and cork-polymer chars, respectively. It is likely that the fillers applied in the PVC mixture increased the thermal stability of the mixed wastes [54].

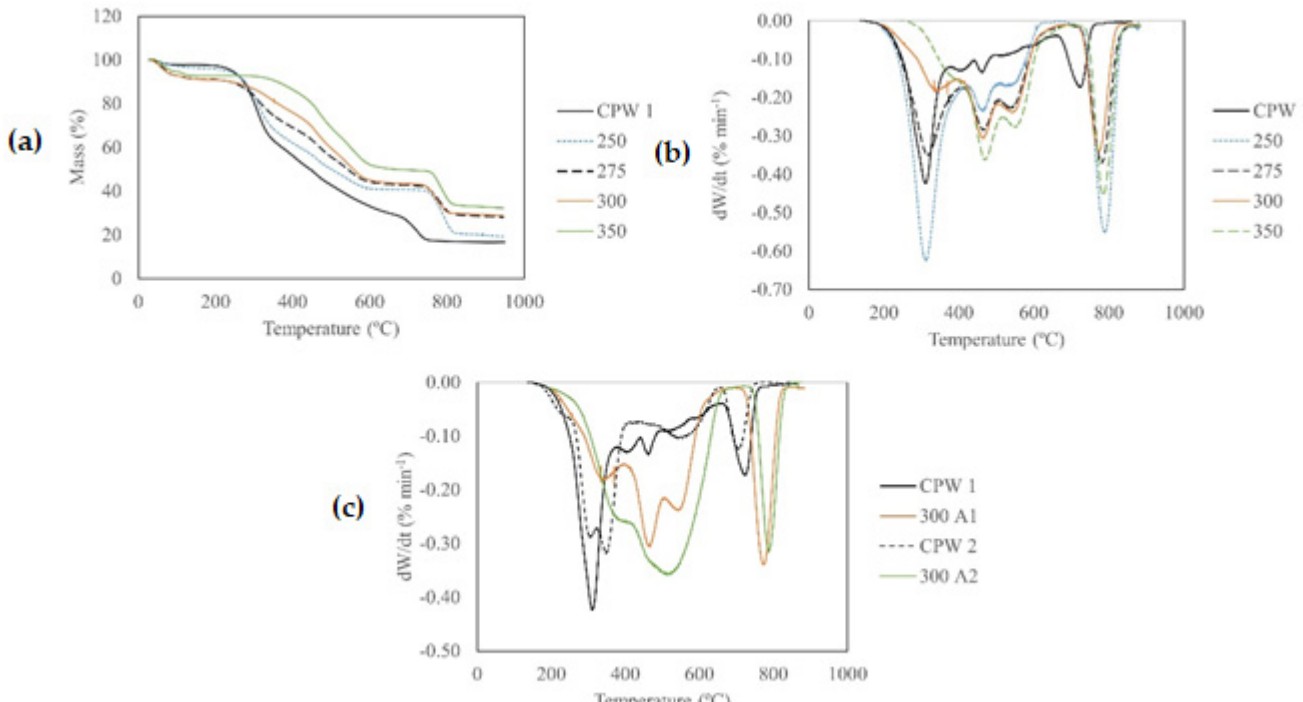

**Figure 5.** Thermal degradation patterns of (**a**) TGA curves of cork-polymer waste 1, (**b**) DTG curves of cork-polymer waste 1, and (**c**) comparative DTG curves of cork-polymer wastes and cork biochars produced at 300 °C.

The ignition and burnout temperatures of cork-polymer wastes and cork-polymer chars were in the range of 260–320 °C and 700–850 °C, respectively (Figure 6). The waste with lower cork content (CPW−1) showed lower ignition temperatures than CPW−2, indicating the presence of higher amounts of oxidable components probably formed during the polymer decomposition [55]. Interestingly, while the ignition temperatures did not change markedly until 300 °C, the burnout temperatures increased significantly after torrefaction at 250 °C and did not change thereafter. The high ignition temperatures indicate that these biochars can be stored without significant risks of spontaneous ignition, but the high burnout temperatures suggest they should be used in high temperature processes such as ceramic kilns that must reach temperatures above 900 °C for the industrial process to take place.

### 3.3.5. FT-IR Analysis

The changes in surface functional groups in the raw material samples and corresponding chars were analyzed by FT-IR. The results are shown in Figure 7.

The broad adsorption band at 3414 cm$^{-1}$ possibly represents the O-H stretching vibration; the absorption bands at 2927 and 2854 cm$^{-1}$ are assigned mainly to asymmetric and symmetric C-H stretching vibrations in the suberin fatty chain, respectively [56]. The bands at 1620 cm$^{-1}$ and 1258 cm$^{-1}$ are attributed to symmetric C=C stretching and C-H rocking of the PVC polymer, respectively [57]. The bands at 1420 cm$^{-1}$, 877 cm$^{-1}$, and 713 cm$^{-1}$ are assigned to C-O stretching and C-O bending vibrations of CaCO$_3$ [58]. CaCO$_3$ is a common filler used in the production of PVC materials [32]. The CaCO$_3$ peaks remained unchanged after torrefaction. The presence of CaCO$_3$ peaks also explains the high ash content in the raw materials.

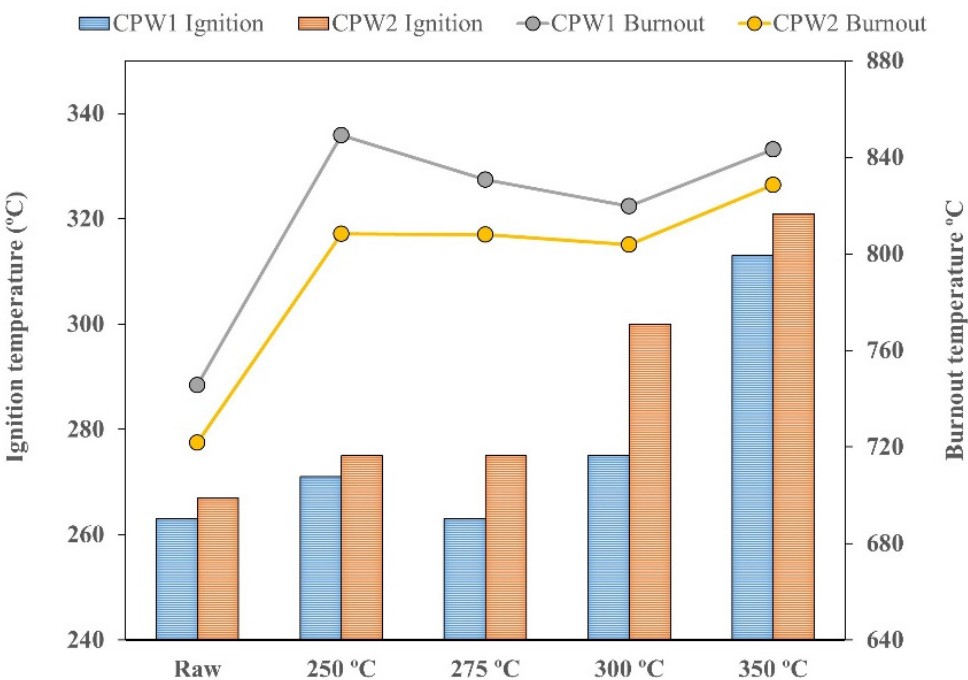

**Figure 6.** The variation in ignition and burnout temperatures of raw cork-polymer wastes and cork-polymer chars produced at 250 °C, 275 °C, 300 °C, and 350 °C.

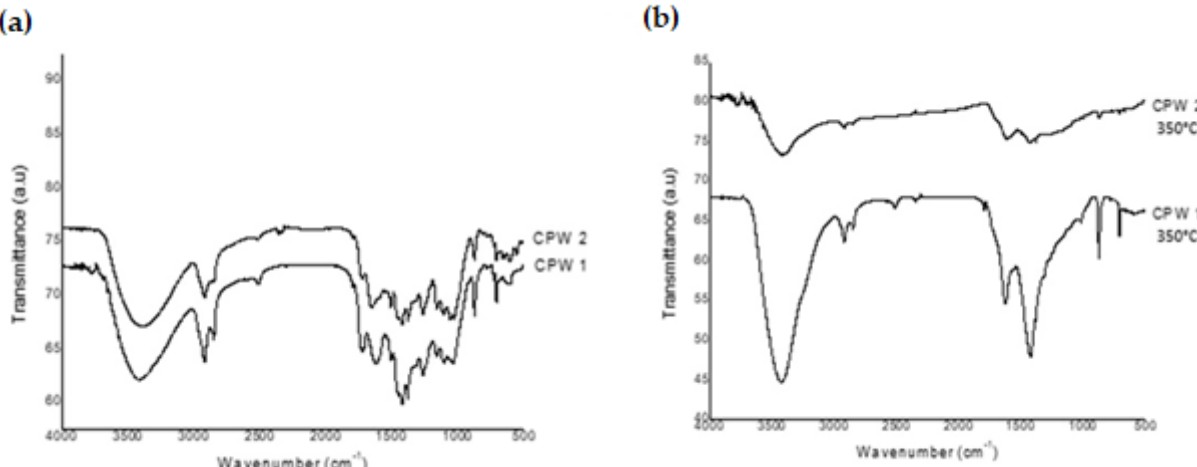

**Figure 7.** FT-IR Spectra of (**a**) CPW−1 and CPW−2 and (**b**) CPW−1 and CPW−2 chars produced at 350 °C and 30 min.

### 3.4. Potential Char Application: Preliminary Adsorption Tests

For the chars whose heating value is not favored by the torrefaction process the material valorization as bio adsorbents, can be an option. In this context, adsorption tests were proposed aiming to evaluate the potential of the different biochars to be used as adsorbents for cationic or anionic species. Removal efficiency for methylene blue (MB), a cationic dye, was evaluated for all chars, before and after activation with concentrated KOH, in experiments with a contact time of a few seconds (instantaneous adsorption) or 1 week (equilibrium adsorption). The results presented in Figure 8 show that the adsorption of MB in the raw materials was higher than 60% for instantaneous adsorption and close to 90% in equilibrium adsorption, reflecting the electrostatic interaction of the cationic dye with the OH or COOH of the lignocellulosic fractions. Torrefaction progressively removed such groups from the biochar surface reducing the adsorption of the cationic dye to values below 20% for instantaneous adsorption and below 70% for equilibrium adsorption. When

the biochars were activated with concentrated KOH, MB equilibrium adsorption increased from 60 to 70% to more than 90% indicating that the activation process removed organic and inorganic components adsorbed in the biochar surface or porous structure, leaving those active sites available for MB adsorption [59].

**(a)**

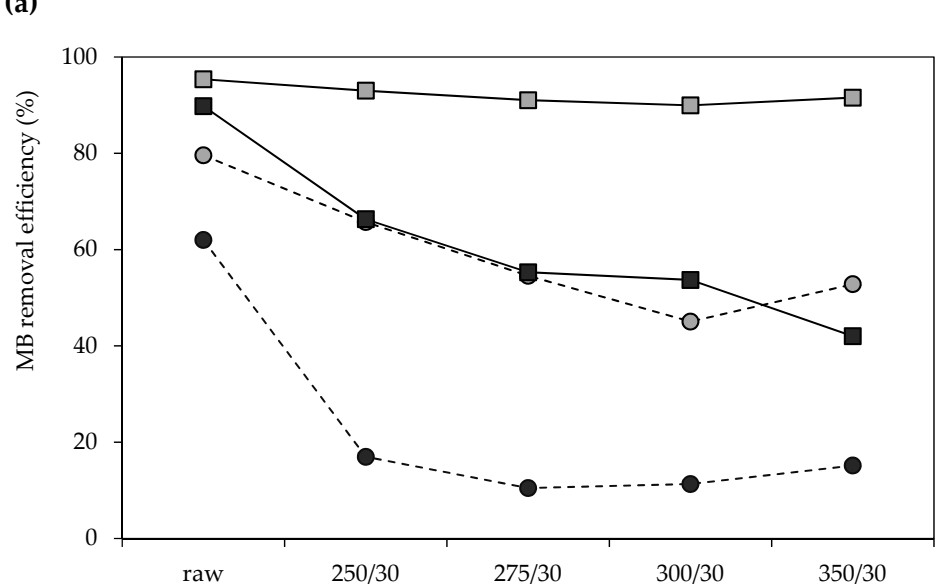

**(b)**

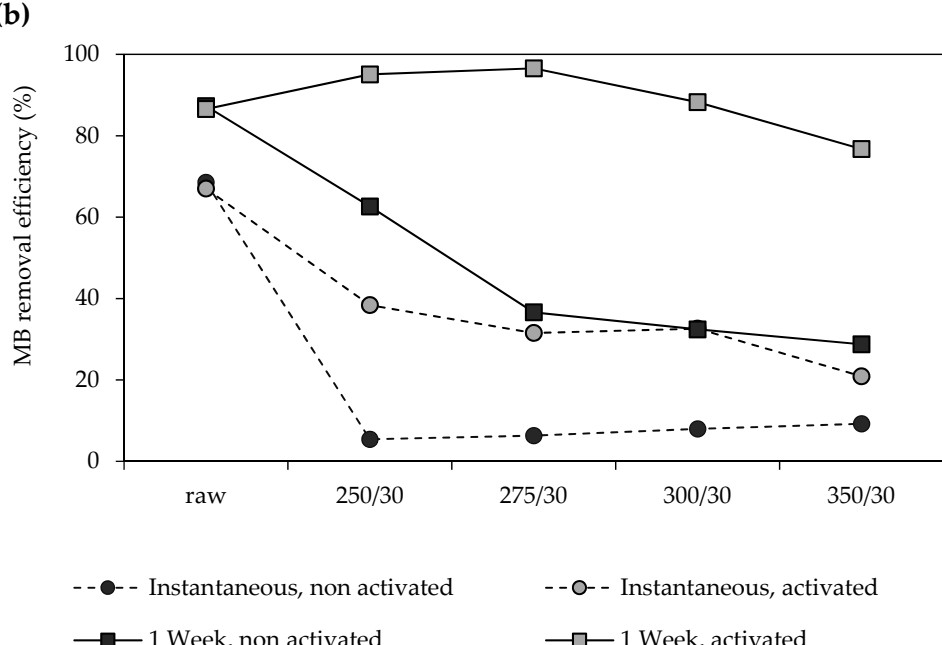

**Figure 8.** MB removal efficiencies for (**a**) CPW-1 and respective chars and (**b**) CPW-2 and respective chars.

The higher adsorption efficiency observed for both biochars (93.0 and 96.6%) is slightly lower than the values of 99.6% obtained for commercial activated carbon [59] and similar to other adsorption studies using activated biochar produced from biomass wastes [60,61].

Regarding the adsorption, both the raw materials and the biochars had a much lower adsorption capacity for MO than for MB because the anionic nature of MO does not favor the electrostatic interaction with negatively charged groups or negative dipoles. Only

after activation with KOH, removal efficiencies higher than 20% were obtained for the biochars derived from CPW-1 and around 45% for the biochars produced from CPW-2, at the temperatures of 250 °C and 300 °C (Figure 9).

**(a)**

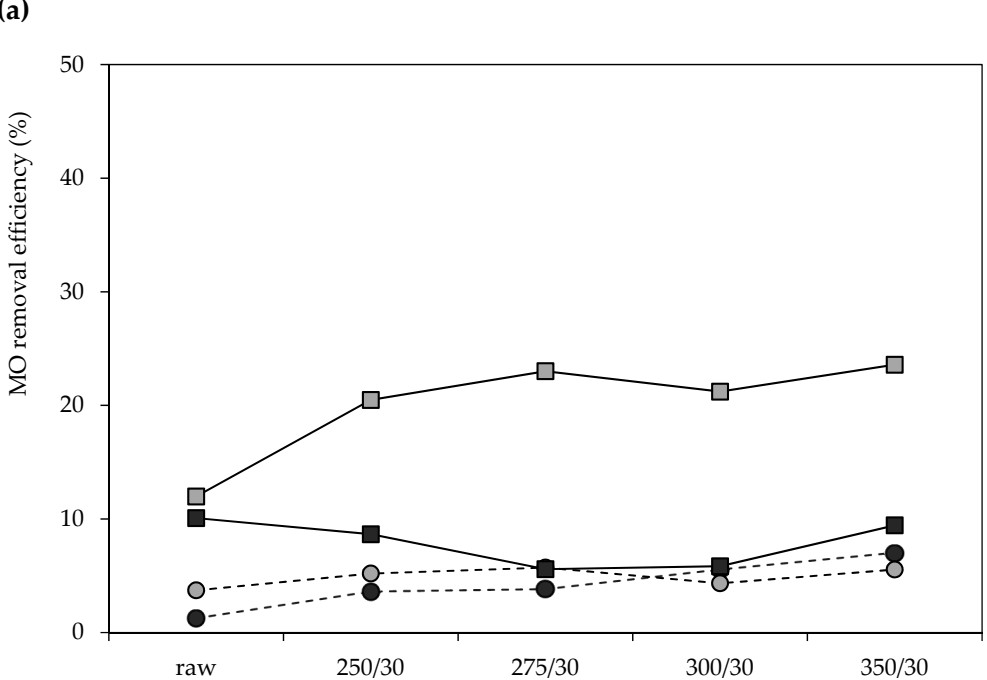

**(b)**

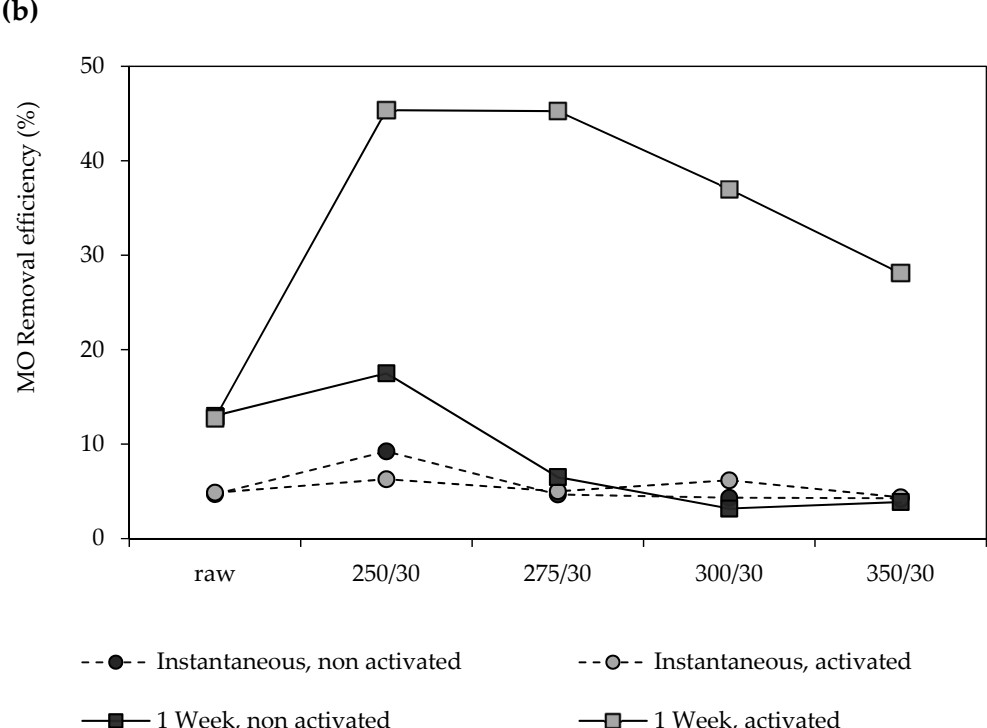

**Figure 9.** MO removal efficiencies for (**a**) CPW-1 and respective chars and (**b**) CPW-2 and respective chars.

These preliminary adsorption results indicate that the CPW-1 biochars can be used for adsorption of cationic dyes, such as MB, but surface adsorption is not enough to ensure

high removal efficiencies thus an activation process is required to promote adsorption in the porous structure.

The poor adsorption of methyl orange (MO), even after activation, indicates that even for torrefaction at higher temperatures, the formation of a carbonaceous structure did not have a direct impact in the adsorption behavior either because the biochar pores are partially occupied by polymer decomposition products that were not removed during activation or because the activation process must be further optimized to achieve higher impact in the biochar adsorption properties. Nonetheless, the highly efficient removal rates for methylene blue suggest that this waste has a good potential to be tested as a low-cost adsorbent to treat some cationic industrial effluents.

## 4. Conclusions

Torrefaction of mixed wastes composed of cork residues and polymers (PVC) had a significant impact on physical and combustible characteristics of these materials. When torrefaction was performed at 250 °C and 300 °C, the waste CPW-2 with higher content of cork biomass was converted to biochars with HHV higher than 18 MJ/kg and less than 1% chlorine, after washing with hot water. Torrefaction also increased apparent density and reduced particle size diameter, properties relevant for energy valorization. The biochars showed higher ignition and burnout temperatures which indicates a higher stability in storage but also requires higher temperatures for complete combustion. Considering some characteristics such as fixed carbon content, HHV, homogeneity, and apparent density, torrefaction between 250 and 350 °C for 30 min appears to convert CPW-2 into biochars with properties suitable for use as fuels.

The composition and high ash content of CPW-1 negatively influenced the fuel properties of the produced biochars, therefore alternative valorization as low-cost adsorbents were evaluated for both wastes and the corresponding biochars.

Both the raw wastes and the derived biochars showed high adsorption capacity (>90%) for cationic dye (MB), after activation with concentrated KOH. Further studies to elucidate adsorption kinetics and thermodynamic equilibria are needed to fully validate this potential valorization pathway.

Globally, torrefaction combined with hot water washing enables the conversion of these wastes into dense biochars with low chlorine content, with potential applications as fuels or adsorbents, thus creating alternative possibilities for promoting circular economy for end-of-life cork products.

**Supplementary Materials:** The following supporting information can be downloaded at: https://www.mdpi.com/article/10.3390/environments9080099/s1, Figure S1: Refuse derived fuel composed of cork and polymer wastes used in the torrefaction tests; Figure S2: Appearance of the chars obtained at different temperatures and residence times; Figure S3: Energy yield and energy density of biochars compared to raw wastes.

**Author Contributions:** Conceptualization, A.L. and M.G.; Data curation, A.L., C.N., A.S. and M.G.; Formal analysis, A.L., C.N. and R.P.; Investigation, A.L. and M.G.; Methodology, A.L. and C.N.; Resources, P.B. and M.G.; Supervision, M.G.; Writing—original draft, A.L.; Writing—review and editing, C.N., A.S. and M.G. All authors have read and agreed to the published version of the manuscript.

**Funding:** This work was supported by national funds through the Fundação para a Ciência e Tecnologia, I.P.P (Portuguese Foundation for Science and Technology) by the project (UIDB/04077/2020-2023 and UIDP/04077/2020-2023) of Mechanical Engineering and Resource Sustainability Center—METRICs, by the project UIDB/00239/2020 of Forest Research Centre—CEF, and by the project UIDB/05064/2020 of VALORIZA—Research Centre for Endogenous Resource Valorization.

**Data Availability Statement:** The data presented in this study are available on request from the corresponding author.

**Acknowledgments:** The authors would like to acknowledge financial support by FCT—Fundação para a Ciência e a Tecnologia within the R&D Units METRICs (UIDB/04077/2020-2023 and UIDP/

04077/2020-2023), CEF (UIDB/00239/2020), and VALORIZA (UIDB/05064/2020). Andrei Longo acknowledges the project AmbWTE–POCI 01-247-FEDER-039838 for the research grant and Ali Sen acknowledges support from FCT through a research contract (DL 57/2016).

**Conflicts of Interest:** The authors declare no conflict of interest.

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
