# Peer review of "Torrefaction Upgrading of Heterogenous Wastes Containing Cork and Chlorinated Polymers"

_environments, doi:10.3390/environments9080099_

Round 1

Reviewer 2 Report

This article from the purpose of the research, the research process and the obt-ained research result perspective may indicate the logic mentality is relatively clear, the research method has the innovation, the research significance is profound.The review comments are as follows: 1 Carbonization as the absence of oxygen heat treatment process, this study m-ixture sample is placed in a covered porcelain crucibles using muffle furnace heating experimental, the accuracy of the experimental results is questionable

2 Carbonization conditions improvements the physical and chemical properties of the torrefied materials, and additional proof methods need to be added to make the torrefied materials more convincing as high combustible and adsorbent materials.
